# Population structure of community-acquired extended-spectrum beta-lactamase producing *Escherichia coli* and methicillin resistant *Staphylococcus aureus* in a French region showed no difference between urban and rural areas

**Adrien Biguenet**[1,2]*, **Xavier Bertrand**[1,2], **Marilou Bourgeon**[3], **Dossi Carine Gnide**[4], **Houssein Gbaguidi-Haore**[1,2], **Céline Slekovec**[1,2]

1 Université de Franche-Comté, UMR-CNRS 6249 Chrono-Environnement, Besançon, France, 2 CHU de Besançon, Hygiène Hospitalière, Besançon, France, 3 CHU de Besançon, Centre de Ressources Biologiques - Filière Microbiologique, Besançon, France, 4 CHU de Besançon, Bioinformatique et Big Data Au Service de La Santé, Besançon, France

* abiguenet@chu-besancon.fr

## Abstract

Antimicrobial resistance is a global health issue and extended-spectrum β-lactamase producing *Escherichia coli* (ESBL-Ec) and methicillin-resistant *Staphylococcus aureus* (MRSA) are of particular concern. Whole genome sequencing analysis of isolates from the community is essential to understand the circulation of those multidrug-resistant bacteria. Our main objective was to determine the population structure of clinical ESBL-Ec and MRSA isolated in the community setting of a French region. For this purpose, isolates were collected from 23 sites belonging to 6 private medical biology laboratories in the Bourgogne-Franche-Comté region. One hundred ninety ESBL-Ec and 67 MRSA were sequenced using the Illumina technology. Genomic analyses were performed to determine the bacterial typing, presence of antibiotic resistance genes, metal resistance genes as well as virulence genes. Analysis showed that ST131 was the major ESBL-Ec clone circulating in the region, representing 42.1% of the ESBL-Ec isolates. The $bla_{CTX-M}$ genes represented 98% of $bla_{ESBL}$ with the majority being $bla_{CTX-M-15}$ (53.9%). MRSA population consisted of mainly of CC8 (50.7%) and CC5 (38.8%) clonal complexes. Interestingly, we found a prevalence of 40% of the zinc resistance gene *czrC* in our MRSA population. We observed no differences in our ESBL-Ec or MRSA populations between urban and rural areas in our French region, suggesting no impact of population density or rural environment.

## Introduction

Multidrug-resistant bacteria (MDRB) are a major public health problem, responsible for increased hospitalization, morbidity and mortality [1]. Two MDRB, *Escherichia coli* and

**Data Availability Statement:** All relevant data are within the paper and its Supporting information files.

**Funding:** This study was funded by the CHU of Besançon. The funders had no role in study design, data collection and analysis, decision to publish, or preparation of the manuscript.

**Competing interests:** The authors have declared that no competing interests exist.

*Staphylococcus aureus*, account for much of the burden of multi-drug resistant bacterial infections, which caused 30,000 deaths in Europe in 2015 [2]. Both these MDRB were initially described in the hospital setting but have therefore spread in the community. Methicillin-resistant *Staphylococcus aureus* (MRSA) was first described in 1961 [3]. By the early 1980s, MRSA had spread worldwide and in some countries was responsible for the majority of *S. aureus* infections in hospitals [4]. MRSA infections initially affected the elderly, children and immunocompromised hospitalized patients. Moreover, in the 1990s, community-acquired MRSA (CA-MRSA) clones emerged and spread widely in the United States, infecting healthy individuals with no history of hospitalization [5]. In France, the main hospital-acquired MRSA (HA-MRSA) clones were ST8 and ST5, while ST80 was mainly community-acquired [6, 7].

Gram-negative multidrug-resistant bacteria and in particular extended-spectrum β-lactamase (ESBL)-producing *Enterobacterales* are a growing public health problem in Europe [8]. The first ESBLs described in 1983 were derived from TEM and SHV enzymes and were produced by *Enterobacterales* responsible for hospital outbreaks. ESBLs carrying $bla_{CTX-M}$ genes were described in France in 1991 in hospitals, but then spread massively into the community. *Escherichia coli* producing ESBL (ESBL-Ec) ST131 is now the most important community clone in Europe and worldwide [9]. It has been observed that most of the hospital-acquired ST131 infections are acquired in the community [10].

In France, a laboratory network monitors antimicrobial resistance in the community [11], but the genomic data available to characterize circulating clones in France are rather rare and outdated [6]. The community setting is composed of rural and urban areas. The One Health approach suggests that the spread and mode of acquisition of MDRB may differ between urban and rural settings [12, 13]. However, the available studies on this particular topic were conducted in Africa or South-East Asia and do not reflect European lifestyles [14–16].

Our main objective was to perform an epidemiological and genomic study of ESBL-Ec and MRSA isolates responsible for community-acquired infections in Bourgogne Franche-Comté region (France). The secondary objective was to compare these multi-resistant strains between rural, semi-urban and urban contexts.

## Methods

### Study design

The study was approved by the local ethic committee before the study commencement considering that no patients data were collected, and samples were anonymised. We conducted a prospective multicenter study including 23 sampling sites belonging to 6 medical biology laboratories in the Bourgogne-Franche-Comté region (France). Laboratories sent to a central laboratory (in the University Hospital of Besançon) all isolates identified as ESBL-Ec or MRSA collected in the first three months of 2019 and 2021 from clinical samples of community patients (only one isolate per patient). Isolates were then categorized according to their origin (urban area >100,000 inhabitants, semi-urban area between 20,000 and 100,000 inhabitants and rural area if < 20,000 inhabitants). The authors did not have access to any information that could identify individual participants during or after data collection.

### Microbiological analysis

Presumptive ESBL-Ec and MRSA were cultured on chromogenic agar plate (URI4 urine, bio-Mérieux, Marcy-L'Etoile, France) and Muller-Hinton agar (Mast Diagnostic, Amiens, France), respectively. Identification was confirmed by MALDI-TOF MS (Microflex LT, Bruker Daltonik GmbH, Bremen, Germany) according to the manufacturer's recommendations (log ≥ 2 and repeatability ≥ 3). Production of ESBL for *E. coli* and resistance to methicillin for MRSA

were checked according to EUCAST 2022 guidelines. Determination of antibiotic resistance to other clinically used antibiotics was also performed according to EUCAST 2022 guidelines.

## Genome sequencing and assembly, antibiotic and metal resistance, virulence factors and typing

Bacterial DNA was extracted using QIAGEN DNeasy UltraClean Microbial Kit (Qiagen, Hilden, Germany) according to the manufacturer's recommendations. The extracted DNA was sequenced on an Illumina NextSeq (150-bp paired-ends reads and a coverage of > 80X). The read sequences are available in the NCBI BioProject PRJNA975705. Raw reads were trimmed using Trimmomatic v0.39 [17] and assembled with SPAdes v3.13.0 [18]. Resistance genes were identified using ResFinder v4.0 [19] by KMA method [20] for enzymatic resistances and using AMRFinderPlus v3.1 [21] to identify point mutations for antibiotic resistance. Virulence genes were identified using VirulenceFinder v2.0 [22, 23] and VFDB [24]. Multi-locus sequence typing (MLST) was performed with pyMLST v2.1.2 (https://github.com/bvalot/pyMLST) (using Achtman and Enright schemes, for *E. coli* and *S. aureus*, respectively). For ESBL-Ec, the virulence genes were classified according to their function as adhesin, toxin and iron acquisition; phylogroup was determined using ClermonTyping v3 [25]; the search for plasmid, chromosomal, or unclassifiable location of ESBL-encoding genes was performed using PlaScope v1.3.1 [26]. For MRSA, metal resistance genes were searched using AMRFinderPlus v3.1; the *spa* and SCC*mec* cassette typing were performed using *spa*Typing [27] and SCC*mec*Finder [28], respectively. All analyses performed had a minimum identity percentage of 90% and a minimum coverage length of 80%.

## Phylogenetic trees—Core genome MLST (cgMLST)

Core genomes of ESBL-Ec and MRSA isolates were determined using pyMLST by aligning genes present from existing schemes deposited on the cgMLST nomenclature server (https://www.cgmlst.org/ncs). Two maximum-likelihood tree were constructed with a GTR + CAT module using FastTree 2.1 [29]. Trees were rooted with the genomes of *E. fergusonii* (GCF_013892435.1) and *S. schweitzeri* (GCF_900636685.1) for ESBL-Ec and MRSA, respectively. Trees were visualized and annotated with iTOL v6.6 [30].

## Statistics

Fisher's exact test or Chi2 test were used for categorical variables and the Wilcoxon-Mann-Whitney test or Kruskal-Wallis test for continuous variables as appropriate. The p values for multiple comparisons were adjusted using the Holm-Bonferroni method. For all analyses, a p value <0.05 was considered statistically significant. Analyses were performed using R studio.

## Results

### *Escherichia coli*—Population structure and antibiotic susceptibility

Among the 190 ESBL-Ec included, phylogroup B2 was predominant with 62.2% of the isolates, phylogroups D and A were represented by 12.1% and 10.0% of the isolates, respectively. We identified 42 different STs among which ST131 predominated (42.1%), the other four most frequently represented STs were ST10, ST69, ST88, and ST1193 with 6.8%, 5.3%, 5.3% and 5.3% of the isolates, respectively. We observed no difference between 2019 and 2021 for phylogroups (p = 0.36), sequence type (p = 0.153) and for the ESBL gene (p = 0.092) in our *Escherichia coli* strains. The population structure of ESBL-Ec is shown in Fig 1 and the phenotypic resistance results displayed in Fig 2. Eighty ESBL-Ec strains (42.1%) were co-resistant to cefotaxime,

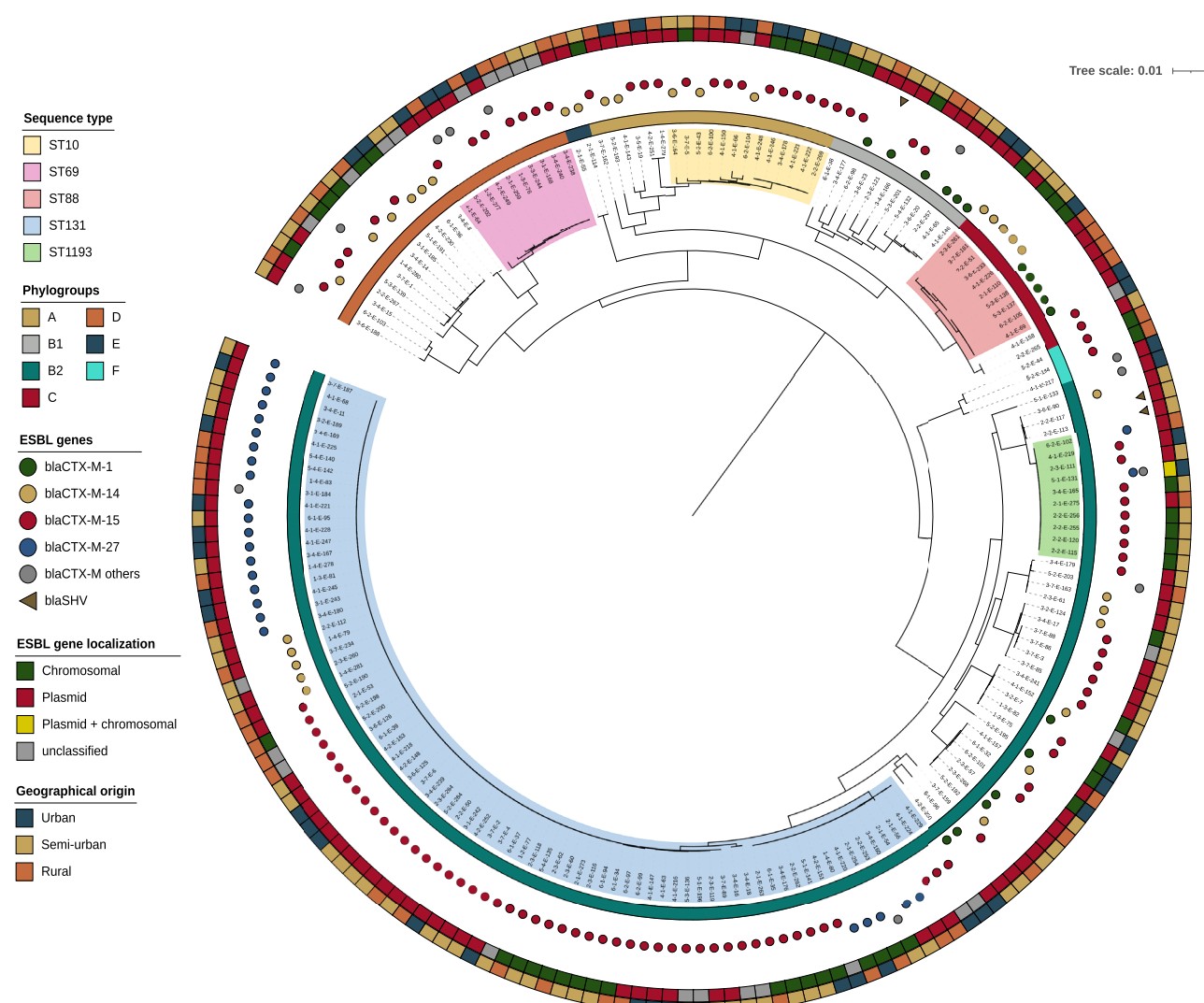

**Fig 1. Phylogenetic tree of 190 clinical ESBL-producing *Escherichia coli* strains isolated in a French region generated by approximately-maximum-likelihood and rooted on *Escherichia fergusonii*.** The major STs are represented by a specific color on the dendrogram, as defined in the legend. The innermost colored ring indicates the phylogroup, while the outermost colored ring indicates the geographical origin of the sample.

ciprofloxacin, and cotrimoxazole. ST131 strains were more resistant to gentamicin (p = 0.01) and to ciprofloxacin (p <0.001) than the strains of other STs.

### *Escherichia coli*—Antibiotic resistance genes

The major ESBL-encoding genes were $bla_{CTX-M-15}$ (53.9%), $bla_{CTX-M-14}$ (16.2%), $bla_{CTX-M-27}$ (13.6%), and $bla_{CTX-M-1}$ (8.9%). One strain carried both $bla_{CTX-M-55}$ and $bla_{CTX-M-27}$. Three isolates harbored $bla_{SHV-12}$ (1.6%) and none $bla_{TEM}$ ESBL. Additionally, $bla_{OXA-1}$ gene was detected in 33.2% (n = 63) of the isolates which were more resistant to piperacillin-tazobactam (44.4%) than isolates without $bla_{OXA-1}$ (6.3%) (p<0.001). The distribution of identified antibiotic resistance genes as well as mutations in QRDR are shown in Fig 2E and 2F, respectively. It should be noted that no carbapenemase-encoding genes was retrieved and that one ST744 strain carried *mcr* 1.1 gene.

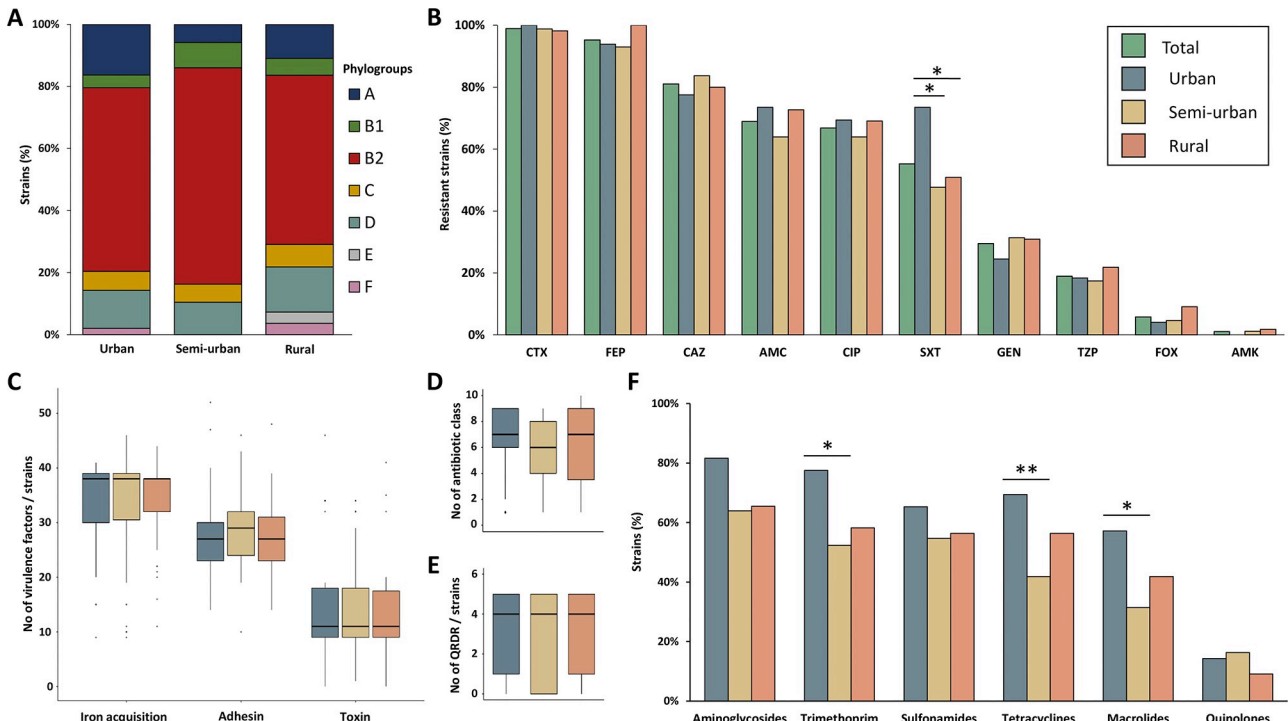

**Fig 2. Comparison of 190 clinical ESBL-producing *Escherichia coli* strains isolated in a French region according to their geographical origin. A**. Distribution of strains according to their phylogroup. **B**. Phenotypic resistance of the strains. **C**. Distribution of the number of virulence genes per strain. **D**. Distribution of the number of antibiotic classes with at least one detected gene per strain. **E**. Distribution of the number of quinolone resistance determining regions (QRDR) per strain. **F**. Proportion of strains with acquisition of at least one antibiotic resistance gene per strain. Significant differences are indicated by asterisks (*: p < 0.05; **: p < 0.01). CTX, cefotaxime; FEP, cefepime; CAZ, ceftazidime; AMC, amoxicillin/clavulanic acid; CIP, ciprofloxacin; SXT, trimethoprim/sulfamethoxazole; GEN, gentamicin; TZP, piperacillin/tazobactam; FOX, cefoxitin; AMK, amikacin.

## *Escherichia coli*—Location of ESBL genes in plasmid and chromosome contigs

We found that 60.7% and 26.7% of ESBL-encoding genes were located on plasmids or on chromosomes, respectively. The location of 12.6% of $bla_{ESBL}$ (19 $bla_{CTX-M-15}$ and 5 $bla_{CTX-M-14}$) could not be determined. The unclassified $bla_{ESBL}$ contigs were smaller than chromosomal (p < 0.001) or plasmid (p = 0.03) contigs containing $bla_{CTX-M-14}/bla_{CTX-M-15}$. The location differed depending on the type of ESBL (p<0.001), with 42.9% (36/84) of $bla_{CTX-M-15}$ and 38.5% (10/26) of $bla_{CTX-M-14}$ being located on chromosomal contigs, whereas $bla_{CTX-M-1}$ was not found in chromosomal contigs. Some ST (ST38, ST10, ST1193 ST88, and ST131) had a high proportion of chromosomal ESBL genes (100%, 58.3%, 55.6%, 40% and 26.1% respectively). The chromosomal or plasmidic location of the ESBL gene had no impact on the number of non-ESBL resistance genes, nor mutations conferring resistance.

## *Escherichia coli*—Comparison between the three areas

ESBL-Ec isolates were collected in urban (n = 49, 26%), semi-urban (n = 86, 45%) and rural (n = 55, 29%) areas. No differences were observed in terms of phylogroup distribution (p = 0.302), number resistance genes (p = 0.25), number of mutated QRDRs (p = 0.847), or distribution of virulence factors (p = 0.847) (Fig 2). Resistance to cotrimoxazole was more frequent in urban than in semi-urban areas (p = 0.011) or rural areas (p = 0.038). We identified

67 strains (35.3%) with resistance genes for macrolides, tetracyclines and trimethoprim (Fig 2). The strains did not belong to specific STs. The proportion of these co-resistant strains was higher in urban areas (24/49) than in semi-urban areas (18/86) (p = 0.002). This difference was not significantly found between urban and rural areas (21/55) (p = 0.267) or between semi-urban and rural areas (p = 0.051).

### *Staphylococcus aureus*—Population structure

Among the 67 MRSA strains included, we identified nine different STs clustered into six clonal complexes (CCs) (S1 Fig). CC8 and CC5 were largely predominant with 34 (50.7%) and 26 (38.8%) isolates, respectively. *spa* typing showed a predominance of *spa* t008 associated to CC8 and t777 associated to CC5 with, respectively, 17 (25.4%) and 18 (26.9%) strains (Fig 3). All of our MRSA strains carried the *mecA* gene. The SCC*mec* type IV cassette was the most frequent (68.6%) followed by the type VI cassette (29.9%) and by the type V cassette (1.5%). CC8 carried SCC*mec* type IV (n = 33) or SCC*mec* type V (n = 1) cassettes. CC5 carried SCC*mec* type VI or type IV for 6 and 20 strains, respectively. We observed no difference between 2019 and 2021 for the complex clonal (p = 0.107) or the SCCmec cassette (p = 0.096) in our MRSA population.

### *Staphylococcus aureus*—Resistance to antibiotics and virulence

*blaZ* gene was present in 54 strains (80.1%) without linkage to CC. The proportion of acquired resistance genes and/or mutations conferring resistance is shown in Fig 4. Two strains (ST8 t121 and ST88) expressed the *lukF* and *lukS* genes responsible for the production of Panton-Valentine leukocidin.

### *Staphylococcus aureus*—Metal resistance

All isolates carried *copA* gene as well as *ars* operon that encode copper and arsenic resistance, respectively. Twenty-eight strains (41.8%) possessed the zinc resistance gene *czrC* (encoding a heavy metal translocating P-type ATPase), for which we identified two alleles with a nucleotide homology of 78%. Twenty-one strains carried the *czrC_1* gene (KF593809.1) and seven strains carried the *czrC_2* gene (CP030605.1). One CC8 t008 strain carried the *mer* operon for mercury resistance and the *cad* operon for cadmium resistance. The *czrC_1* gene were present in the SCC*mec* type VI cassette of all CC5 t777 strains (n = 18). The *czrC_1* gene was also found in the SCC*mec* type IV cassette of two CC8 t008 strains and in the SCC*mec* type VI cassette of one CC5 t2379 strain. The presence of the *czrC_1* gene in the SCC*mec* type IV cassette was associated with the additional presence of the *ccrA4* and *ccrB4* genes between the *orfX* region and the IS431 insertion sequence. The sequences *ccrB4-ccrA4-czrC_1* had a nucleotide homology of more than 99% between CC5 t777 and CC8 t008. The 7 *czrC_2* genes were present in the SCC*mec* type IV cassette in six CC8 t008 strains and in one CC398 strain.

### *Staphylococcus aureus*—Comparison between the three areas

MRSA strains were collected in urban (n = 14, 20.9%), semi-urban (n = 33, 49.3%) and rural areas (n = 20, 29.8%) areas. The *blaZ* gene was less frequent in rural areas (10/20) than in semi-urban areas (32/33, p<0.001). The distribution of zinc resistance genes was similar in the three zones (p = 0.86). We did not observe any difference in the distribution of CC (p = 0.57), number of virulence genes (p = 0.64) and number of resistance genes (p = 0.26) (Fig 4).

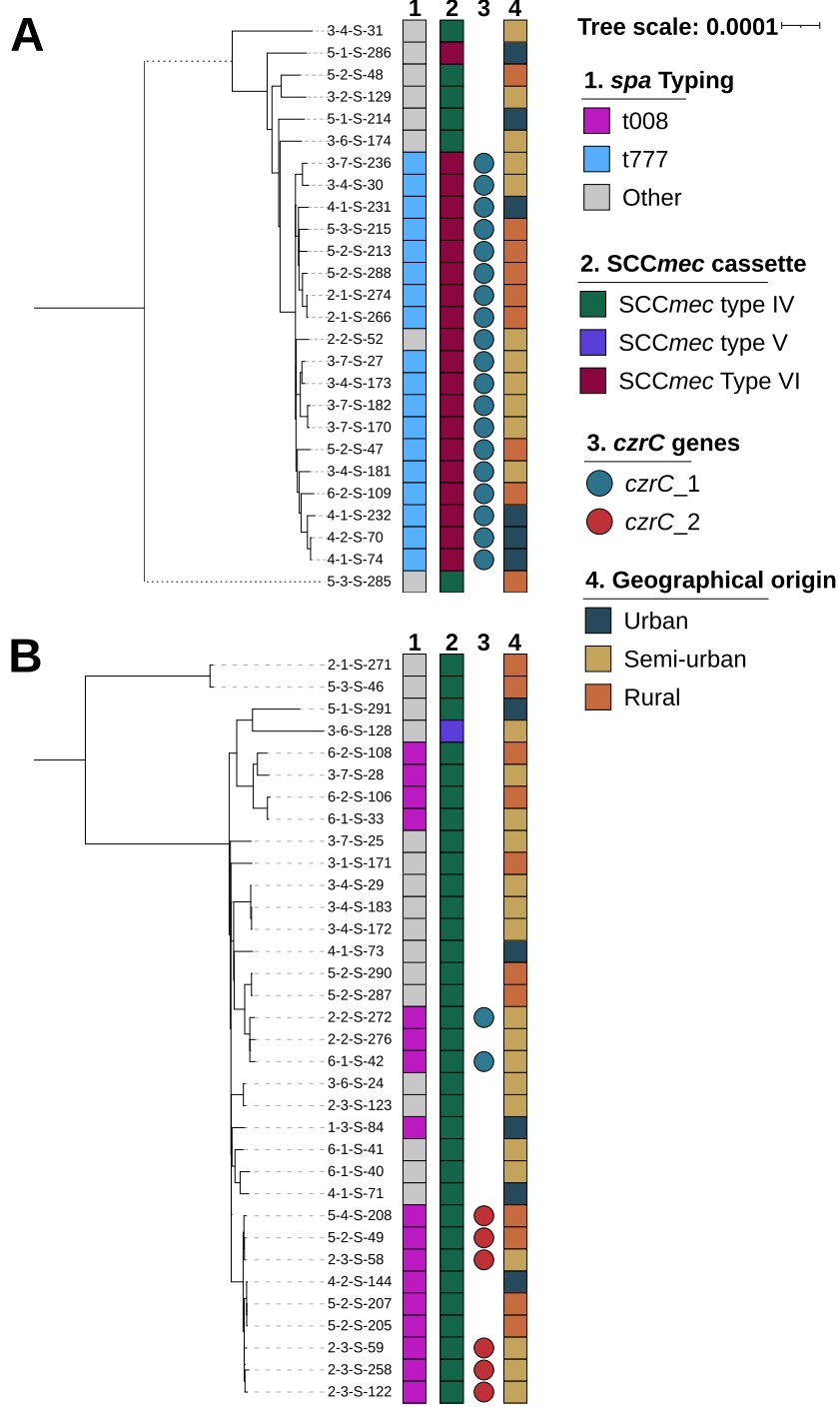

**Fig 3. Phylogenetic tree of CC5 and CC8 methicillin-resistant *Staphylococcus aureus* isolates in a French region in 2019–2021.** Trees were generated by approximately-maximum-likelihood and rooted with *Staphylococcus schweitzeri*. **A**. CC5 methicillin-resistant *Staphylococcus aureus*. **B**. CC8 methicillin-resistant *Staphylococcus aureus*.

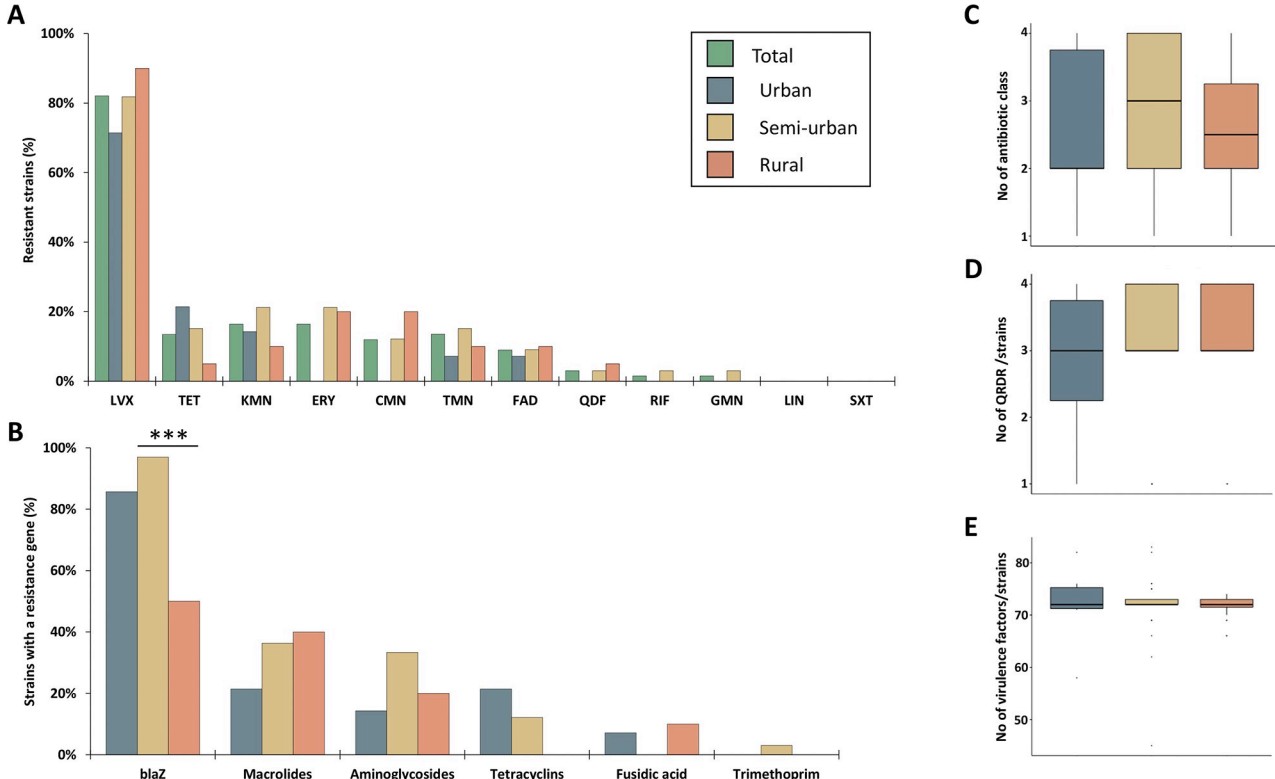

**Fig 4. Comparison of 67 clinical methicillin-resistant *Staphylococcus aureus* strains isolated from a French region according to their geographical origin. A**. Phenotypic resistance of the strains. **B**. Proportion of strains with acquisition of at least one antibiotic resistance gene per strain. **C**. Distribution of the number of antibiotic classes with at least one detected gene per strain. **D**. Distribution of the number of quinolone resistance determining regions (QRDR) per strain. **E**. Distribution of the number of virulence genes per strain. Significant differences are indicated by asterisks (***: p < 0.001). LVX, levofloxacin; TET, tetracycline; KMN, kanamycin; ERY, erythromycin; CMN, clindamycin; TMN, tobramycin; FAD, fusidic acid; QDR, quinupristin/dalfopristin; RIF, rifampicin; GMN, gentamicin; LIN, linezolid; SXT, trimethoprim/sulfamethoxazole.

## Discussion

In this community-based study on the scale of a French region, whole genome sequencing of a significant set of ESBL-Ec and MRSA revealed that ESBL-Ec population is largely dominated by the ST131 clone and $bla_{CTX-M}$ genes and that MRSA belongs to 2 major clones whose origin is probably the hospitals. National surveillance data for the region [31] suggests that our collection is representative since it represents approximately 60% of ESBL-Ec and MRSA isolated during the study period. These results are consistent with most European surveillance studies that include genotyping [32–34]. We also found similar population structures of ESBL-Ec and MRSA between, urban, semi-urban or rural areas suggesting no impact of population density or rural environment on the epidemiology of these MDRB.

Although $bla_{ESBL}$ genes are mostly borne on plasmids (especially for $bla_{CTX-M-1}$), chromosomal integration is frequent for $bla_{CTX-M-14}$ and $bla_{CTX-M-15}$ genes. The localization of 12.6% of our $bla_{ESBL}$ gene could not be identified, probably due to the size of the contigs. Chromosomal integration of $bla_{CTX-M}$ has already been observed [35] and could be beneficial for the propagation of the Ec-ESBL as in *Klebsiella pneumoniae* [36].

Co-resistance associated with ESBL genes is a major problem for patient treatment. More than 40% of the strains present in our study are co-resistant to fluoroquinolones and cotrimoxazole. These resistances are particularly problematic for the management of pyelonephritis and

male urinary tract infections, which require the prescription of carbapenems or intravenous injection of antibiotics several times a day [37]. Several studies have revealed that the $bla_{OXA-1}$ gene is responsible for increased MICs of piperacillin/tazobactam (TZP) and may be responsible for increased mortality [38, 39]. We identified a prevalence of 33% in our community ESBL-Ec strains, associate to a resistance to TZP of 20% in our Ec-ESBL population. This resistance rate is similar to that of French hospital strains [40]. Probabilistic monotherapy with TZP in patients at high risk of ESBL could expose to a considerable risk of failure.

We identified in our ESBL-Ec strains a significant increase in co-resistance to cotrimoxazole, tetracyclines and macrolides in urban areas. These co-resistant strains mainly harbored *mph (A)*, *dfrA17* and *tet(A)* as already observed in *E. coli* ST131 [41]. Tetracycline consumption has been identified as a significant cause of ESBL acquisition [42] One hypothesis is that the use of tetracyclines selected the strains resistant to tetracyclines, cotrimoxazole and macrolides.

MRSA isolated in our community share the characteristics of MRSA isolated in French hospital. Indeed, CC5 t777 (New pediatric clone) and CC8 t008 (Lyon Clone) represent together more than 50% of the MRSA isolated in our study. Strains considered as CA-MRSA are rare, with only one strain USA300 (ST8 t121, SCC*mec* IV, PVL+) and one ST88 strain that produced PVL [43]. The USA300 clone is endemic in the United States but remains rare in France. No ST80 PVL-producing strain (European clone) was found in our study although it has been identified as the major CA-MRSA strain in Europe [44]. ST398 has emerged in the last two decades to take an important place in the livestock-associated MRSA population. Nevertheless, only 2 MRSA ST398 strains were identified in our study, in contrast with the high proportion of MSSA ST398 recovered in our region [45].

We identified the presence of zinc resistance genes in more than 40% of our human MRSA population. Zinc resistance was initially associated to LA-MRSA, notably ST398. The high prevalence of *czrC* in our study is surprising given that MRSA ST398 strains [46] represent only 3% of our population. These *czrC*-containing strains can be separated into two distinct populations: CC5 SCC*mec* VI *czrC_1* and CC8 SCC*mec* IV *czrC_2* strains. The presence of *czrC_1* genes in two CC8 t008 SCC*mec* IV strains is likely explained by the integration of a *ccrA4-ccrB4-czrC_1* sequence similar to our CC5 t777 SCC*mec* VI strains. This suggests a high mobilization capacity of the *czrC_1* gene. Recently, a 22% prevalence of *czrC* was observed in ST5 SCC*mec* II strains of human origin in the United States [47]. Our study is the first, to our knowledge, to identify a dissemination of two distinct lineages of MRSA carrying two different *czrC* alleles.

Our study has several limitations. First, we did not recover the origin of the sample, but it is likely that the majority of ESBL-Ec were from urine samples while MRSA were from skin samples. Second, the comparison between urban, peri-urban, and rural areas implies that people have little contact between areas and that they go to their local laboratories for testing, which we were unable to verify. Thirdly, the patients were all living at home at the time of sampling, but we did not have information on hospitalization history, which can explain the carriage of MRSA of hospital origin [48].

In conclusion, our results provide additional data for monitoring the evolution of MDRB in the community sector, in which genomic data are quite rare. Our data confirm previous data. We did not highlight differences between urban and rural strains.

## Supporting information

**S1 Fig. Phylogenetic tree of 67 clinical methicillin-resistant *Staphylococcus aureus* isolated in a French region in 2019–2021.**
(EPS)

**S1 Table. Origin of the studied isolates and genomic results.**
(XLSX)

## Acknowledgments

We would like to thank the following laboratories for their willingness to participate in the study and for sending us the strains: BCLAB, BIOGROUPE LCD, BIOLABUNILABS, CBM25, LPA and MEDILYS. We thank Benoit Valot for software support and Alexandre Meunier for technical support.

## Author Contributions

**Conceptualization:** Xavier Bertrand, Céline Slekovec.

**Data curation:** Adrien Biguenet, Marilou Bourgeon.

**Formal analysis:** Adrien Biguenet.

**Funding acquisition:** Céline Slekovec.

**Methodology:** Houssein Gbaguidi-Haore.

**Software:** Adrien Biguenet, Dossi Carine Gnide.

**Supervision:** Xavier Bertrand.

**Writing – original draft:** Adrien Biguenet, Marilou Bourgeon.

**Writing – review & editing:** Xavier Bertrand, Dossi Carine Gnide, Houssein Gbaguidi-Haore, Céline Slekovec.

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
