## [Decision Letter · Decision Letter 0]

11 Jul 2023

PONE-D-23-15982Population structure of community-acquired extended-spectrum beta-lactamase producing Escherichia coli and methicillin resistant Staphylococcus aureus in a French region showed no difference between urban and rural areasPLOS ONE

Dear Dr. Biguenet,

Thank you for submitting your manuscript to PLOS ONE. After careful consideration, we feel that it has merit but does not fully meet PLOS ONE’s publication criteria as it currently stands. Therefore, we invite you to submit a revised version of the manuscript that addresses the points raised during the review process.

We look forward to receiving your revised manuscript.

Kind regards,

Zhi Ruan, Ph.D.

Academic Editor

PLOS ONE

Journal Requirements:

"This study was funded by the CHU of Besançon (API-CHU 2019)"

Reviewers' comments:

Reviewer's Responses to Questions

**Comments to the Author**

1. Is the manuscript technically sound, and do the data support the conclusions?

Reviewer #1: Yes

Reviewer #2: Partly

2. Has the statistical analysis been performed appropriately and rigorously? 

Reviewer #1: Yes

Reviewer #2: Yes

3. Have the authors made all data underlying the findings in their manuscript fully available?

Reviewer #1: Yes

Reviewer #2: Yes

4. Is the manuscript presented in an intelligible fashion and written in standard English?

Reviewer #1: Yes

Reviewer #2: Yes

5. Review Comments to the Author

Reviewer #1: Antimicrobial resistance is an urgent global public health threat，with high mortality rates due to infections caused by multidrug-resistant bacteria. Understanding the molecular epidemiology of multidrug-resistant bacteria acquired in the community is crucial for effectively preventing and controlling infections caused by these resistant strains.

There are also some major concerns,

1. In line 74, the authors mentioned conducting a prospective multicenter study. It is unclear whether this study underwent ethical review. Moreover, the sampling methods and detailed information regarding the samples have not been specified.

2. In Figure 1, it would be helpful to annotate each circle with specific information to indicate what each circle represents.

3. Do the authors analyze the MLST of the MRSA isolates?

4. Do the authors compare the evolutionary changes of multidrug-resistant bacteria between 2019 and 2021 in the region?

Reviewer #2: Biguenet et al. has performed a population structure/genomic analysis of ESBL producing E. coli and MRSA isolates in France - reporting the prevalence of STs and AMR genes. The authors has also studied the distribution of these characteristics in relation to the region/areas based on population size. In general the study was interesting and important to public health, and the MS was well written. However, the classification of three areas such as urban and rural was quite arbitrary, and not convincing.

Major questions:

1. line 79: population size was not the only factor. other factors such as anthropogenic activities, animal farming, agriculture, or winery would be more relevant to the relationship of isolates under one health approach. So the authors did not know where the patients were from?

2. line 169-170: The association of ESBLs and plasmids was not based on solid methodology, as only short-read sequencing data were used to determine the genetic location of these genes. Multicopy insertion elements may fragment the short-read assemblies into multiple contigs, and reconstruction of ESBL plasmids requires additional experiments, such as long-read sequencing or PCR gap closure.

3. Line 298-304: The lack of clinical data or relevant metadata had weakened the discussion and entire investigation. It was unusual that the origin of these isolates were unknown. Without knowing the source, all the discussion was kind of speculation, even though Ec ST131 tended to be circulated among the communities. Was the study supported by the ethical board?

6. PLOS authors have the option to publish the peer review history of their article (what does this mean?). If published, this will include your full peer review and any attached files.

Reviewer #1: No

Reviewer #2: No

---

## [Author Response · Author response to Decision Letter 0]

25 Aug 2023

Reviewer #1: 

Antimicrobial resistance is an urgent global public health threat, with high mortality rates due to infections caused by multidrug-resistant bacteria. Understanding the molecular epidemiology of multidrug-resistant bacteria acquired in the community is crucial for effectively preventing and controlling infections caused by these resistant strains. 

There are also some major concerns,

1. In line 74, the authors mentioned conducting a prospective multicenter study. It is unclear whether this study underwent ethical review. Moreover, the sampling methods and detailed information regarding the samples have not been specified.

The study was approved by the local ethic committee before the study commencement considering that no patients data were collected and samples were anonymised. This was added in the text line 74. Information regarding samples were specified in Lines 77-80.

2. In Figure 1, it would be helpful to annotate each circle with specific information to indicate what each circle represents.

The colors are listed in the figure legend from the inner circle to the outer circle.

3. Do the authors analyze the MLST of the MRSA isolates?

We chose to analyse complex clones (CC) rather than sequence types (ST) due to the small size of our population. The results of the MLST analysis are available in the supplementary data.

4. Do the authors compare the evolutionary changes of multidrug-resistant bacteria between 2019 and 2021 in the region?

We observed no difference between 2019 and 2021 for phylogroups (p = 0.36), sequence type (p = 0.153) and for the ESBL gene (p = 0.092) in our Escherichia coli strains. For Staphylococcus aureus, we observed no difference for the complex clonal (p = 0.107) or the SCCmec cassette (p = 0.096). This was added in the text line 135 and line 204.

Reviewer #2: 

Biguenet et al. has performed a population structure/genomic analysis of ESBL producing E. coli and MRSA isolates in France - reporting the prevalence of STs and AMR genes. The authors has also studied the distribution of these characteristics in relation to the region/areas based on population size. In general the study was interesting and important to public health, and the MS was well written. However, the classification of three areas such as urban and rural was quite arbitrary, and not convincing.

Major questions:

1. Line 79: population size was not the only factor. other factors such as anthropogenic activities, animal farming, agriculture, or winery would be more relevant to the relationship of isolates under one health approach. So the authors did not know where the patients were from?

No patient data were collected, so we agree with the reviewer that the location of the laboratory is a distant proxy of patient environment and anthropogenic activities. This limitation is addressed in the discussion section.

2. Line 169-170: The association of ESBLs and plasmids was not based on solid methodology, as only short-read sequencing data were used to determine the genetic location of these genes. Multicopy insertion elements may fragment the short-read assemblies into multiple contigs, and reconstruction of ESBL plasmids requires additional experiments, such as long-read sequencing or PCR gap closure.

We agree that short reads do not allow total plasmid reconstruction. We have used PlaScope here, a tool to predict whether the contigs from short read data in our fasta file are from chromosome, plasmid or unclassified location. PlaScope uses Centrifuge and a custom database of chromosome and plasmid sequences to classify the contigs. PlasScope is not able to reconstruct the plasmid or predict if 2 contigs are from the same or different plasmids.

Other tools are also available to distinguish between plasmid and short read contigs:

- Robertson J and Nash J. RFPlasmid: predicting plasmid sequences from short-read

data using machine learning. Microbial Genomics. 2018. DOI: 10.1099/mgen.0.000683

- van der Graaf-van Bloois et al. MOB-suite: software tools for clustering, reconstruction and typing of plasmids from draft assemblies. Microbial Genomics. 2021. DOI: 10.1099/mgen.0.000206 (nécessaire ces 2 citations ?)

3. Line 298-304: The lack of clinical data or relevant metadata had weakened the discussion and entire investigation. It was unusual that the origin of these isolates were unknown. Without knowing the source, all the discussion was kind of speculation, even though Ec ST131 tended to be circulated among the communities. Was the study supported by the ethical board?

We agree with the reviewer that the lack of clinical data is the major limitation of our study and we underlined this point in the discussion. However, we have collected all consecutive ESBLE and MRSA isolates in a large number of private community laboratories, and it is very likely that our results reflect population structures of ESBLE and MRSA in our region. Moreover, such data are not frequently available in the community sector.

The study was approved by the local ethic committee before the study commencement considering that no patients data were collected and samples were anonymised

---

## [Decision Letter · Decision Letter 1]

2 Nov 2023

Population structure of community-acquired extended-spectrum beta-lactamase producing Escherichia coli and methicillin resistant Staphylococcus aureus in a French region showed no difference between urban and rural areas

PONE-D-23-15982R1

Dear Dr. Biguenet,

We’re pleased to inform you that your manuscript has been judged scientifically suitable for publication and will be formally accepted for publication once it meets all outstanding technical requirements.

Kind regards,

Zhi Ruan, Ph.D.

Academic Editor

PLOS ONE

Additional Editor Comments (optional):

Reviewers' comments:

Reviewer's Responses to Questions

**Comments to the Author**

1. If the authors have adequately addressed your comments raised in a previous round of review and you feel that this manuscript is now acceptable for publication, you may indicate that here to bypass the “Comments to the Author” section, enter your conflict of interest statement in the “Confidential to Editor” section, and submit your "Accept" recommendation.

Reviewer #1: All comments have been addressed

Reviewer #2: All comments have been addressed

2. Is the manuscript technically sound, and do the data support the conclusions?

Reviewer #1: Yes

Reviewer #2: Partly

3. Has the statistical analysis been performed appropriately and rigorously? 

Reviewer #1: Yes

Reviewer #2: Yes

4. Have the authors made all data underlying the findings in their manuscript fully available?

Reviewer #1: Yes

Reviewer #2: Yes

5. Is the manuscript presented in an intelligible fashion and written in standard English?

Reviewer #1: Yes

Reviewer #2: Yes

6. Review Comments to the Author

Reviewer #1: (No Response)

Reviewer #2: All the questions have been addressed in the revised manuscript. An ethical statement has been included in the methods

7. PLOS authors have the option to publish the peer review history of their article (what does this mean?). If published, this will include your full peer review and any attached files.

Reviewer #1: **Yes: **Tian Jiang

Reviewer #2: No

---

## [Editor Report · Acceptance letter]

6 Nov 2023

PONE-D-23-15982R1 

Population structure of community-acquired extended-spectrum beta-lactamase producing *Escherichia coli* and methicillin resistant *Staphylococcus aureus* in a French region showed no difference between urban and rural areas 

Dear Dr. Biguenet:

I'm pleased to inform you that your manuscript has been deemed suitable for publication in PLOS ONE. Congratulations! Your manuscript is now with our production department. 

Kind regards, 

on behalf of

Dr. Zhi Ruan 

Academic Editor

PLOS ONE